# Metabolomics in Preclinical Drug Safety Assessment: Current Status and Future Trends

**DOI:** 10.3390/metabo14020098

**Published:** 2024-01-31

**Authors:** Fenna Sillé, Thomas Hartung

**Affiliations:** 1Center for Alternatives to Animal Testing (CAAT), Department of Environmental Health Sciences, Johns Hopkins Bloomberg School of Public Health and Whiting School of Engineering, Johns Hopkins University, Baltimore, MD 21205, USA; fsille1@jhu.edu; 2CAAT-Europe, University of Konstanz, Universitätsstraße 10, 78464 Konstanz, Germany

**Keywords:** metabolomics, toxicity, safety, drug development, adverse outcome pathways

## Abstract

Metabolomics is emerging as a powerful systems biology approach for improving preclinical drug safety assessment. This review discusses current applications and future trends of metabolomics in toxicology and drug development. Metabolomics can elucidate adverse outcome pathways by detecting endogenous biochemical alterations underlying toxicity mechanisms. Furthermore, metabolomics enables better characterization of human environmental exposures and their influence on disease pathogenesis. Metabolomics approaches are being increasingly incorporated into toxicology studies and safety pharmacology evaluations to gain mechanistic insights and identify early biomarkers of toxicity. However, realizing the full potential of metabolomics in regulatory decision making requires a robust demonstration of reliability through quality assurance practices, reference materials, and interlaboratory studies. Overall, metabolomics shows great promise in strengthening the mechanistic understanding of toxicity, enhancing routine safety screening, and transforming exposure and risk assessment paradigms. Integration of metabolomics with computational, in vitro, and personalized medicine innovations will shape future applications in predictive toxicology.

## 1. Introduction

Metabolomics, the global study of metabolites in a biological system, has emerged as a powerful technology for elucidating mechanisms of toxicity and improving preclinical drug safety assessment. By providing a direct snapshot of biochemical activity and a functional readout of cellular responses, metabolomics data can help unravel adverse outcome pathways, strengthen extrapolation of preclinical findings, and aid regulatory decision making regarding potential safety liabilities. This article reviews key opportunities where metabolomics approaches can inform and transform current paradigms in preclinical toxicology evaluation during drug development. The utility of metabolomics for elucidating adverse outcome pathways, incorporating human exposure and mechanistic information, and enhancing predictive toxicity evaluation is discussed [1]. Quality assurance considerations, which are essential for regulatory acceptance of metabolomics data, are highlighted. Overall, metabolomics represents a promising opportunity to strengthen the scientific basis of safety assessment and improve preclinical hazard characterization.

Adverse outcome pathways (AOPs) represent a conceptual framework for organizing existing toxicological knowledge to support chemical risk assessment and regulatory decision making [2]. AOPs link molecular initiating events to downstream key events at increasing levels of biological organization, culminating in an adverse outcome relevant to risk assessment. By capturing the current understanding of causal connections within plausible chains of events leading to toxicity, AOPs provide a mechanistic rationale to aid toxicity prediction. Metabolomics provides direct insights into biochemical activity and perturbations that occur downstream of gene expression changes. Consequently, metabolomics data can help anchor omics-derived AOPs to adverse outcomes and strengthen causal links between key events. Metabolomics data can help anchor omics-derived adverse outcome pathways (AOPs) to adverse outcomes and strengthen causal links between key events. Metabolomics represents an effective approach for identifying additional AOPs that may be occurring in parallel with an already assigned AOP, as well as providing physiological support and expanding knowledge of those AOPs relevant to a particular exposure [3]. Metabolomics can measure changes in endogenous metabolites involved in a wide variety of biochemical pathways and associate those changes with exposure to specific chemicals and adverse biological outcomes. In the context of ecological risk assessment, omics datasets, including metabolomics, enable a more precise definition of the molecular initiating event (MIE) and the selection of biomarkers relevant for assessing effects and/or exposure [4]. Omics data provide gene- and pathway-level readouts that can be used for selecting measurable and relevant biomarkers. For example, metabolomics has been used to describe the AOPs of silver nanoparticles [5], selenium as it relates to brain toxicity [6], and the pharmaceutical spironolactone [3]. Metabolomics can also be used to find similarities between biological responses to different chemicals to facilitate chemical grouping for read-across of adverse events. Metabolomics is a powerful tool used to investigate the responses of metabolite profiles in organisms exposed to various substances. For instance, a study applied metabolomics to investigate the responses of metabolite profiles in zebrafish exposed to acetochlor and butachlor, two substances that could disrupt the thyroid and sex steroid endocrine systems in zebrafish [7]. Metabolomics can give insights into toxicity pathways and mechanisms, e.g., metabolomics is an emerging approach used to identify and discover metabolic biomarkers, providing a series of metabolic signatures that elucidate the pathological changes in diseases, including Parkinson’s disease [8]. This technology links various metabolic molecular mechanisms to neuronal activity alterations, protein changes or genetic mutations, and mitochondrial dysfunction. Metabolomics also serves the identification of early, sensitive, causal biomarkers, e.g., it revealed biomarkers of drug-induced liver injury that provided mechanistic clues [9,10]. The concept of bridging in vitro and in vivo effects using metabolomics to evaluate an adverse outcome pathway (AOP) for neuronal mitochondrial toxicity across multiple models is supported by several studies [11,12].

By supporting AOP development, metabolomics strengthens predictive toxicity assessment based on elucidating mechanisms of chemical perturbations. Metabolomics promises to play a key role in elucidating the connection between chemical exposure and human health. The exposome encompasses the totality of environmental exposures an individual experiences across a lifetime [13]. Chemical exposome research utilizing metabolomics can shed light on how such exposures interplay with genes and lifestyle to influence pathogenesis. For example, metabolomics analysis of urine can detect metabolic changes following chemical exposures [14]. Furthermore, the ability to acquire metabolomics snapshots of individuals represents a powerful tool for elucidating connections between chemical exposures, metabolic disturbances, and disease outcomes in human population studies [15]. By aiding more accurate exposure assessment and linking exposures to biological impacts, metabolomics research can transform our understanding of environmental influences on human health.

Metabolomics approaches are increasingly being applied in toxicology and safety pharmacology to elucidate mechanisms of toxicity, identify predictive safety biomarkers, and enhance routine screening [16,17,18,19,20,21]. By detecting endogenous metabolite alterations indicating biochemical perturbations, metabolomics provides functional readouts of pathology that link traditional toxicity endpoints to mechanistic pathways. For example, metabolomics has shown utility for revealing mechanisms underlying organ toxicity of drugs and environmental chemicals [22]. Furthermore, incorporating metabolomics into repeat-dose toxicology studies can provide additional weight of evidence and insights beyond standard parameters [23]. Metabolomics can also inform in vitro to in vivo extrapolation (IVIVE) for safety assessment by anchoring in vitro assays to meaningful apical endpoints [24]. Overall, metabolomics shows promise for strengthening mechanistic understanding in safety assessment. The Organisation for Economic Co-operation and Development (OECD) has also recognized the potential of metabolomics in regulatory toxicology, paving the way for applications in hazard (or adverse outcome) identification, chemical grouping to inform biologically based read-across of toxicity, identifying metabolic points-of-departure, and potentially other areas [25].

Quality assurance needs are key for the utility of metabolomics to support drug development and regulatory decisions. While metabolomics holds tremendous potential for informing preclinical drug safety evaluation, expanded use will require robust quality assurance and demonstration of reliability for regulatory decision making [26,27]. Key needs include standardized protocols for sample preparation and data acquisition, metabolite identification validation, ongoing quality control using reference materials, transparency in data analysis workflows, and structured reporting formats to support interpretation [28,29]. Furthermore, fit-for-purpose qualification of specific metabolomics approaches will be necessary for regulatory acceptance. Collaborative efforts between metabolomics researchers, toxicologists, and regulators to establish best practices and evaluation frameworks will enable the translation of metabolomics innovations into improved safety testing paradigms.

In summary, metabolomics approaches are poised to transform preclinical drug safety assessment by elucidating adverse outcome pathways, incorporating exposome and mechanistic information, and enhancing predictive toxicity evaluation. However, realizing the full potential of metabolomics will require continued focus on quality assurance and qualification to support regulatory decision making. Overall, metabolomics represents a promising opportunity to strengthen the scientific basis of safety assessment and improve preclinical hazard characterization.

### 1.1. Adverse Outcome Pathways vs. Pathways of Toxicity

A broader assessment of metabolites, as enabled by metabolomics, lends itself to the identification of affected pathways in toxicological research. Adverse outcome pathways (AOPs) and pathways of toxicity (PoTs) represent related conceptual frameworks for organizing toxicological knowledge to support risk assessment. While complementary, they have been defined with distinct perspectives [30]. AOPs were introduced in the field of ecotoxicology and focus on linking mechanistic data to apical hazards or outcomes relevant to regulatory decision making [2]. AOPs connect a molecular initiating event to subsequent key events using causal relationships to provide a plausible narrative explaining how chemical perturbation leads to an adverse effect [31]. They play a critical role in the design of integrated testing strategies [32,33], also known as integrated approaches to testing and assessment (IATA) [34]. However, AOPs do not require a full understanding or representation of every step in the pathway.

In contrast, PoTs aim to provide a molecular definition of all processes mediating adverse outcomes induced by toxicants in a given cellular context [35] (Figure 1). PoTs intend to map the necessary and sufficient pathways eliciting toxicity to support predictive hazard assessment and move away from animal testing. Furthermore, PoTs emphasize understanding dose-dependent dynamics and temporal relationships.

While AOPs represent a very structured organization of scientific knowledge from chemical exposure to population effects, PoTs use experimental models and multi-omics approaches to link chemical exposure to cellular effects.

While AOPs offer a pragmatic framework to incorporate existing data for regulatory purposes, PoTs represent a more ambitious goal of mapping all key toxicity pathways as a new paradigm for safety assessment. Metabolomics approaches, by detecting endogenous changes indicative of biochemical impacts, can help strengthen the development of both AOPs and PoTs. However, fully realizing the vision of PoT-based toxicology will require major advances in high-throughput systems approaches and pathway mapping tools. Ultimately, AOPs and PoTs both seek to leverage mechanistic knowledge but differ in scope and application. They are the critical link from the exposome to the understanding of disease [36,37]. The critical role of evidence-based approaches in their development and validation has been discussed elsewhere [38,39].

### 1.2. Metabolomics to Understand Human Exposure Contributions to Disease and Treatment Efficacy

Metabolomics has emerged as a powerful technology for gaining insights into how human exposures impact metabolic pathways and influence disease outcomes. By profiling global changes in endogenous small molecules, metabolomics provides functional biomonitoring data that complements traditional exposure assessments [40] (Figure 2).

Some key applications of metabolomics for elucidating exposure contributions to human disease include the following:

**Characterizing Effects of Drug Exposures**: Metabolomics has been widely applied in pharmacology and toxicology to delineate drug exposure impacts on metabolic pathways [18,41]. This ranges from the development of biomarkers and early toxicity tests to preclinical drug development and increasingly regulatory toxicology. The global study of metabolism via metabolomics has significant implications for pharmacologic science. By capturing the overall physiological status of an organism, metabolomics studies offer a comprehensive understanding of how drugs influence the body’s biochemistry at a macro level. Metabolomics also has potential applications in clinical pharmacology, where it can provide insights into drug responses and safety profiles.

**Elucidating Dietary and Nutritional Influences**: As diet represents a major source of diverse small molecule exposures, metabolomics holds promise for elucidating nutritional influences on health. This plays a role as a covariable of pathogenesis and treatment efficacy or even allows tailored nutrition as co-treatment [42].

**Understanding Lifestyle Factor Impacts:** Various lifestyle factors have been investigated using metabolomics approaches. Tobacco smoke exposure, alcohol consumption, physical activity, and sleep patterns are lifestyle exposures that can be addressed with metabolomics [43] and possibly impact drug efficacy.

**Elucidating Mixture Effects:** Real-world exposures involve complex mixtures of chemicals and other stressors; most patients receive multiple drugs. Metabolomics provides an opportunity to evaluate the integrated biological impacts of co-exposures [44]. The technology often allows the detection of mixture-associated effects not evident from individual exposures.

In summary, metabolomics serves as a critical link between human exposures and resultant biochemical alterations that influence disease outcomes and treatment efficacy, also enabling personalized medicine [45]. By offering insight into exposure-induced disturbances in metabolic pathways, metabolomics holds great promise to elucidate environmental contributions to pathogenesis and reveal molecular mechanisms underlying clinical outcomes.

## 2. Metabolomics Data Quality Assurance in Preclinical Drug Safety Assessment

The emergent field of metabolomics is providing new opportunities to gain insights into drug safety and toxicity through comprehensive profiling of metabolic changes. However, as with any new technology, there are quality issues and sources of variability that need to be addressed to ensure metabolomics data can be reliably utilized in preclinical drug safety workflows [21].

There are several potential sources of variability that can impact the quality of metabolomics data:

**Biological variability between study subjects and across study groups**: Biological variability between study subjects and across study groups can significantly impact the quality of metabolomics data. This variability can be due to individual differences in metabolism, diet, lifestyle, and other factors. It is important to account for this variability in the study design and statistical analysis to improve the quality of metabolomics data [46,47,48].

Remedies: Careful experimental design with sufficient replication and randomization of study subjects to account for individual variability; blocking on key factors known to influence metabolism (sex, age, genotype, etc.); and use of linear mixed-effects models in data analysis to model biological variability.

**Sample collection and handling procedures:** Sample collection and handling procedures are crucial in metabolomics studies. Inappropriate sample collection or storage can result in high variability, interferences with instrumentation, or degradation of metabolites. Therefore, it is important to follow specific standard operating procedures (SOPs) for the collection, preparation, and storage of metabolomics samples [49,50].

Remedies: Develop and validate standard operating procedures (SOPs) for each sample type; use consistent sampling time points relative to study conditions; standardize processing workflows (e.g., stabilization and aliquoting); employ automated sample handling where possible to reduce errors; and validate storage conditions to ensure sample integrity.

**Metabolite extraction methods:** Metabolite extraction methods can also impact the quality of metabolomics data. The extraction efficiency and repeatability can be highly variable across protocols, tissues, and chemical classes of metabolites. Therefore, the choice of extraction method should be carefully considered based on the sample type and the metabolic compounds of interest [51,52].

Remedies: Assess extraction efficiency/variability for method and sample type; use multiple complementary extraction protocols for broad coverage if needed; include technical replicates in extractions to monitor process variability; and select methods minimizing metabolite losses or artifacts.

**Analytical instrumentation:** Gas chromatography (GC), liquid chromatography (LC), and mass spectrometry (MS), as well as nuclear magnetic resonance (NMR) plays a crucial role in metabolomics studies. However, instrumental drifts, such as fluctuations in retention time and signal intensity, can pose challenges, particularly in large untargeted metabolomics studies. Therefore, quality control (QC) samples are often used to improve the validity of these studies [23,43].

Remedies: Employ quality control samples and reference materials to monitor performance; set acceptance criteria for instrument stability (retention times and internal standards); use randomization in analytical run sequences to minimize batch effects; and perform periodic instrument maintenance and calibration.

**Data acquisition parameters and protocols:** These parameters and protocols can also influence the quality of metabolomics data. For example, the number of variables (peaks, chemical shifts, ions, etc.) can be very large, while the number of observations remains usually low in metabolomics datasets. Therefore, appropriate data analysis methods are needed to handle this high dimensionality [43].

Remedies: Perform injections of standards to optimize acquisition parameters upfront; use a randomized run order and bracketed injections of QC samples; monitor QC metrics (peak intensities, shapes, and blanks) to ensure stable performance; employ automated data acquisition workflows to minimize analyst-induced variability; and share methods/protocols to improve transparency and reproducibility.

**Data processing pipelines for peak identification, quantification, and statistical analysis:** data processing pipelines for peak identification, quantification, and statistical analysis are crucial for the quality of metabolomics data. Various strategies for technical improvements in metabolomics have been aimed at raising data quality, including methods to reduce the effects of sample matrices, maximize sensitivity during detection, separate isomers, reduce ion suppression, and improve instrument performance [43].

Remedies: Leverage quality metrics to filter poor quality data features; use multiple algorithms for identification/quantification to reduce errors; incorporate replicate injections and pooled QC samples to assess process variability; and employ statistical methods suited for high dimensionality data.

In conclusion, small variations in any step of the metabolomics workflow, from experimental design to data analysis, can introduce noise and bias, leading to irreproducible results [23]. Adhering to quality assurance protocols throughout the metabolomics workflow allows critical assessment and improvement in factors contributing to variability, enhancing reproducibility. A comprehensive discussion of best practices strengthens confidence in utilizing metabolomics for safety assessment. Careful experimental design and standardization of procedures are critical. Quality control (QC) procedures, such as the use of reference materials, quality control samples, and evaluation of technical replicates, are essential for monitoring variability and ensuring the day-to-day performance of metabolomics assays [24]. Quality assurance (QA) protocols, including staff training, method validation, and standard operating procedures, establish the robustness and reliability of the overall metabolomics workflow.

Adhering to QA/QC protocols promotes intra- and inter-laboratory reproducibility, allowing metabolomics data to be compared across studies and sites [22]. Ongoing community efforts have established some consensus QA/QC practices and reporting standards for metabolomics through interlaboratory studies, published guidelines, and the development of reference materials [21,22,53]. Viant et al. and Kirwan et al. suggested reporting standards [54,55]. Wider adoption of these standards will improve the quality and consistency of metabolomics data utilized in preclinical drug safety assessment.

## 3. Role of Metabolomics in Toxicology

Metabolomics has emerged as a powerful tool for studying the toxicity and safety of drugs and environmental chemicals. By comprehensively analyzing the small molecule metabolites in biological systems, metabolomics provides insights into metabolic pathways disrupted by toxicants and can identify early biomarkers of toxicity. Some examples of how metabolomics has advanced toxicology research include providing mechanistic insights into compound toxicity or early metabolic biomarkers of toxicity.

An example of metabolomics studies providing mechanistic insights into compound toxicity is the study of acetaminophen (APAP). APAP is a commonly used analgesic and antipyretic drug, but it poses a major risk of liver injury when taken in excess of the therapeutic dose. The hepatotoxicity is initiated by the formation of a reactive metabolite N-acetyl-p-benzoquinone imine (NAPQI), which depletes cellular glutathione and forms protein adducts on mitochondrial proteins. This leads to mitochondrial oxidative and nitrosative stress, resulting in liver injury [56]. Metabolomics studies of acetaminophen toxicity in rodents [57,58,59] showed alterations in bile acid metabolism and increased oxidative stress markers prior to overt signs of liver injury, providing mechanistic clues to the pathogenesis of acetaminophen hepatotoxicity. Additionally, metabolites related to glutathione metabolism and oxidative stress, such as cysteine-glutathione disulfide and oxidized glutathione, were increased by acetaminophen. Another example of identification is the metabolomic analysis of lung cell tissues from different species exposed to cigarette smoke, which revealed the disruption of glycolysis, the Krebs cycle, choline metabolism, and additive oxidative stress, elucidating the metabolic impact of cigarette smoke on the lungs [60,61,62]. Metabolomics profiling showed decreased glucose and elevated lactate, indicating an inhibition of glycolysis. Additionally, alterations in Krebs cycle metabolites like fumarate and malate were observed, suggesting impairment of mitochondrial function. Phosphocholine levels were increased, and glycerophosphocholine levels decreased in smoke-exposed rat lungs, reflecting disruption of choline metabolism. Markers of oxidative stress like glutathione disulfide were also elevated. The metabolomics data provided a detailed picture of the metabolic pathways perturbed by cigarette smoke in the lungs, including glycolysis dysfunction, oxidative stress, and altered choline metabolism. Similarly, metabolite profiles of HepG2 cells treated with 35 test substances resulted revealed concentration–response effects and patterns of metabolome changes that are consistent for different liver toxicity mechanisms (liver enzyme induction/inhibition, liver toxicity, and peroxisome proliferation) [63]. In an in vitro model of MPP^+^-inducible Parkinsonism, metabolomic flux analysis was able to characterize the cell death of dopaminergic neurons [64].

Examples of identification of early metabolic biomarkers of toxicity using metabolomics include a serum metabolomics study, which identified metabolite biomarkers of liver and kidney toxicity in rats, with some biomarkers detectable prior to changes in classical clinical chemistry markers of tissue injury [65]. Metabolomics analysis showed liver toxins decreased amino acids and increased bile acids in rat serum, while kidney toxicants altered tricarboxylic acid cycle (TCA) metabolites and prostaglandins. Elevations in serum bile acids were detectable earlier than ALT increases with liver toxicants, while TCA cycle changes preceded creatinine increases for kidney toxicants. This demonstrates the potential of metabolomics for discovering early metabolic biomarkers of organ toxicity. Another example is a metabolomic analysis, which discovered biomarkers for early detection of nephrotoxicity. The study found that one of the earliest and strongest metabolic changes induced by the three nephrotoxicants was an increase in urinary excretion of 2-oxoglutarate [66]. Metabolomics profiling was also used to detect early effects of environmental and lifestyle exposure to cadmium in a human population. The study concluded that metabolic profiling has the potential to identify novel biomarkers and molecular signatures of the effects of exposure to many environmental toxicants [14].

In summary, as these and further non-comprehensive examples (Table 1) illustrate, metabolomics has provided toxicologists with a powerful systems biology tool to gain mechanistic insights, discover early biomarkers, and assess interindividual differences in response to toxins. Metabolomics allows a comprehensive analysis of metabolic pathway perturbations from toxicant exposure, bridging the gap between mechanistic research and classical toxicity endpoints. The ability to detect subtle metabolite changes prior to overt signs of organ damage enables metabolomics to identify early biomarkers of toxicity for improved risk assessment. Furthermore, metabolomic analyses can uncover metabolic differences underlying interindividual variability in susceptibility to toxicants. Continued development of metabolomics technologies and databases will further enhance the application of metabolomics in 21st-century predictive toxicology and advance the field of toxicology toward personalized and precision medicine.

## 4. State of the Art of Metabolomics in Toxicology

Metabolomics has emerged as a transformative approach in toxicology (Figure 3). This technology enables us to delve into the biochemical fingerprints that cellular processes leave behind, thereby offering unique insights into the adverse outcome pathways (AOPs) and the underlying mechanisms of toxicity [74,75]. This article aims to provide an overview of the current state of metabolomics in toxicology and discusses its future trends, focusing on its role in preclinical drug safety assessment.

**Addressing the challenges of toxicology in the 21st century** [76]: Metabolomics is ideally positioned as a powerful tool for collecting rich mechanistic information indicating not only the extent of a toxic insult but also its underlying mechanisms. The metabolome of a sample, that is, the concentrations of these metabolites at a given time, can be thought of as a metabolic “fingerprint” representative of the state of the organism at that time.

**Discovering Adverse Outcome Pathways (AOPs)**: Traditional toxicological studies have been largely dependent on empirical observations, often overlooking the biochemical changes that precede visible symptoms. One objective in developing AOPs is to connect biological changes that are relevant to risk assessors to molecular- and cellular-level alterations that might be detectable at earlier stages of chemical exposure. Metabolomics, however, allows for the identification of early biomarkers and signaling pathways that can serve as AOPs, which link the initial molecular interaction of a substance with its final adverse effect [2]. This adds a layer of granularity to toxicological assessments, enabling more precise risk evaluations and interventions. Metabolomics represents an effective approach for not only identifying the presence of additional AOPs that may be occurring in parallel with an already assigned AOP but also for providing physiological support and expanding knowledge of those AOPs relevant to a particular exposure.

**Metabolomics and Human Exposure:** The contribution of metabolomics extends to assessing the biological impact of environmental exposure. By analyzing metabolite profiles, scientists can gauge the influence of various xenobiotics, thus elucidating the exposure contribution to diseases [35,77]. Metabolomics offers a panoramic view of how endogenous metabolites interact with environmental toxins, a step closer to the exposome concept, which represents the totality of human environmental exposures from conception onwards [13]. Recently, the vision of a Human Exposome Project was developed [78,79]: The “Future Directions Workshop—Advancing the Next Scientific Revolution in Toxicology” sponsored by the Basic Research Office, Office of the Under Secretary of Defense for Research and Engineering covers a wide range of topics related to toxicology, including the concept of the Human Exposome. Based on the information available in the document, here is a summary of the vision of a Human Exposome Project: The project envisions mapping and quantifying the myriad of external factors, including chemicals, biological agents, and lifestyle factors, alongside internal processes such as metabolism and inflammation. The project’s goal is to integrate these diverse data sets to create a more holistic picture of how environmental factors interact with genetic predispositions to influence health and disease. At its core, the Human Exposome Project seeks to revolutionize the field of toxicology by shifting the focus from individual toxicants to the complex interplay of various environmental factors. This involves the use of advanced technologies such as high-throughput screening, bioinformatics, and systems biology to analyze and interpret large datasets. The project aims to provide insights into the mechanisms of disease, identify new biomarkers for early disease detection, and inform risk assessment and preventive strategies. It represents a paradigm shift in understanding human health and disease, highlighting the importance of environmental factors in shaping health outcomes.

**Role of Metabolomics in Toxicology:** Beyond preclinical drug safety, metabolomics has applications in mechanistic toxicology, forensic toxicology, and regulatory toxicology. It is instrumental in drug development pipelines, complementing other “omics” technologies like genomics and proteomics [36]. Its potential to generate high-throughput data makes it valuable for large-scale toxicological screenings and in the formation of integrated testing strategies (ITSs) [27].

**Quality Assurance in Metabolomics:** As metabolomics becomes increasingly integrated into regulatory toxicological studies, there is a growing need for quality assurance and standardized protocols. Efforts like Good Cell Culture Practice and Good Read-Across Practice are laying the foundation for methodological harmonization [80,81]. Ensuring data quality and comparability across different laboratories is essential for the credibility and applicability of metabolomics data.

## 5. Future Trends in Metabolomics for Toxicology

Metabolomics is poised to transform the field of toxicology by providing unprecedented insights into the metabolic perturbations caused by toxicants. Advances in computational tools, in vitro models, and personalized medicine will shape the trajectory of metabolomics in toxicology research and regulatory applications. Key future trends include the following (Figure 4):

**Artificial Intelligence in Metabolomics:** The massive quantity and complexity of metabolomics data necessitates advanced computational methods for data interpretation. Machine learning and artificial intelligence tools are rapidly emerging in toxicology [82,83,84] and will glean meaningful biological insights from complex metabolomics data sets. For example, deep learning neural networks can infer properties from similar compounds with test data [85] and integrate multi-omics data and predict toxicity outcomes [86]. Maertens et al. (2017) demonstrated that certain machine learning algorithms for metabolomic network analysis of estrogen-stimulated MCF-7 cells outperform traditional over-representation analysis, quantitative enrichment analysis, and pathway analysis [87]. Unsupervised clustering algorithms can uncover latent data structures to classify toxicants by mechanism of action [88]. As metabolomics studies continue producing expansive, multi-dimensional data, artificial intelligence and big data analytics will become integral to deriving value from metabolomics for toxicity risk assessments.

**Integrating metabolomics with other ‘omics’ technologies:** The combination with omics such as genomics, proteomics, and transcriptomics offers a comprehensive and holistic approach to understanding the biological impact of drugs [89,90]. This multi-omics integration enables a deeper insight into the systemic effects of drug-induced changes, providing a more complete picture of the molecular landscape. Metabolomics, which focuses on the end products of cellular processes, can reveal the functional consequences of genetic and protein alterations caused by drug exposure. By correlating metabolomic data with genomic variations and protein expression patterns, researchers can identify specific biomarkers and mechanistic pathways involved in drug toxicity and efficacy. This integrative approach not only enhances the accuracy of preclinical drug safety assessments but also paves the way for personalized medicine. It allows for the identification of individual susceptibility to drug-induced adverse effects based on a person’s unique molecular profile, thereby improving drug safety and efficacy. Additionally, the combination of these ‘omics’ technologies fosters a more robust understanding of the interplay between different biological layers, ultimately leading to more effective and safer therapeutic interventions.

**Microphysiological Systems:** Combining metabolomics with microphysiological systems (MPSs) [91,92,93], such as organ-on-a-chip or 3D organoid models, offers more physiologically relevant platforms for toxicology studies [94,95]. Metabolomics analysis of lab-on-a-chip devices containing multiple connected microorganisms can provide system-wide insights into compound effects [96]. Moreover, MPSs derived from stem cells allow toxicant screening against specific genotypes, enabling personalized toxicity evaluation [97]. Integrating metabolomics with advanced in vitro models will enhance the prediction of compound toxicity in humans.

**Personalized Toxicology:** The emergence of precision medicine has kindled interest in developing personalized toxicology approaches tailored to individual risk profiles. Metabolomics is poised to enable personalized toxicology by capturing individual metabolic variability in response to drugs and toxicants [98]. For example, metabolomics could identify vulnerable subpopulations at greater risk for adverse effects. Additionally, baseline metabolomics profiles may one day guide more precise toxicological risk assessments and interventions for individuals.

In conclusion, metabolomics is revolutionizing 21st-century toxicology through a symbiotic relationship with computational tools, microphysiological systems, and personalized medicine. As these fields continue advancing, metabolomics will become an indispensable pillar of modern toxicology, unlocking unprecedented insights into xenobiotic perturbations and enabling more predictive, accurate and individualized safety assessments.

## Figures and Tables

**Figure 1 metabolites-14-00098-f001:**
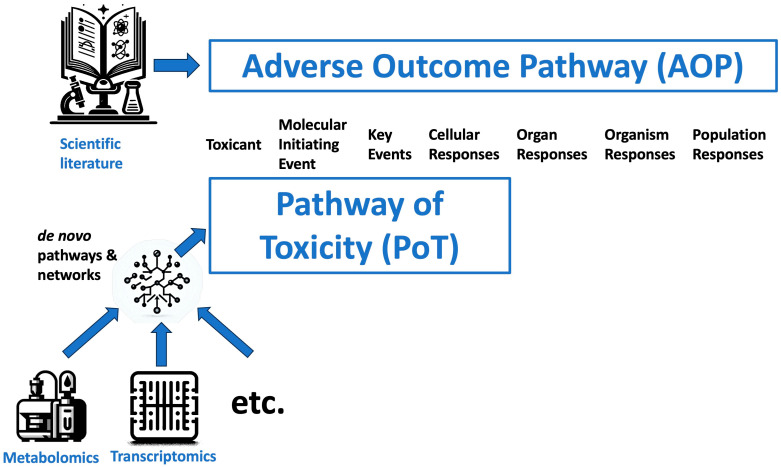
Comparison of concepts of adverse outcome pathways (AOPs) and pathways of toxicity (PoT).

**Figure 2 metabolites-14-00098-f002:**
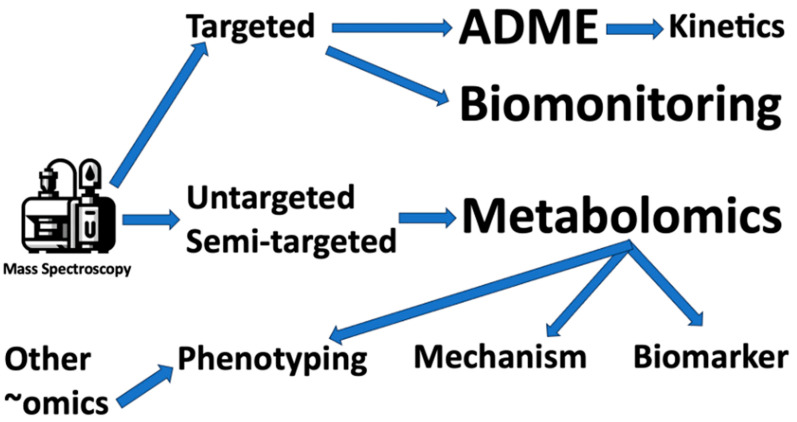
Major uses of mass spectroscopy in toxicology. ADME = adsorption, distribution, metabolism, and excretion.

**Figure 3 metabolites-14-00098-f003:**
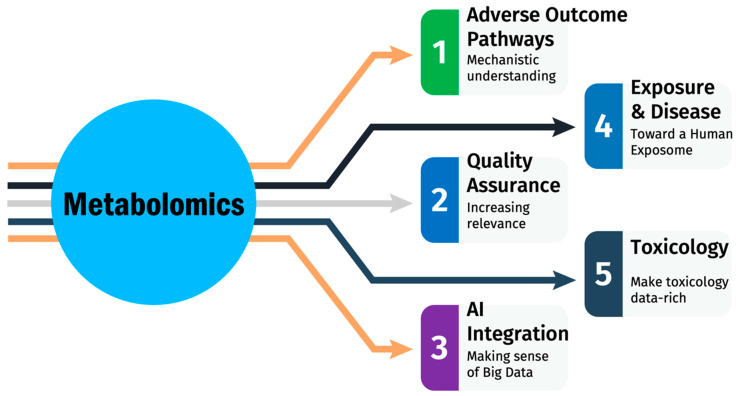
Current trends in metabolomics for drug safety assessments.

**Figure 4 metabolites-14-00098-f004:**
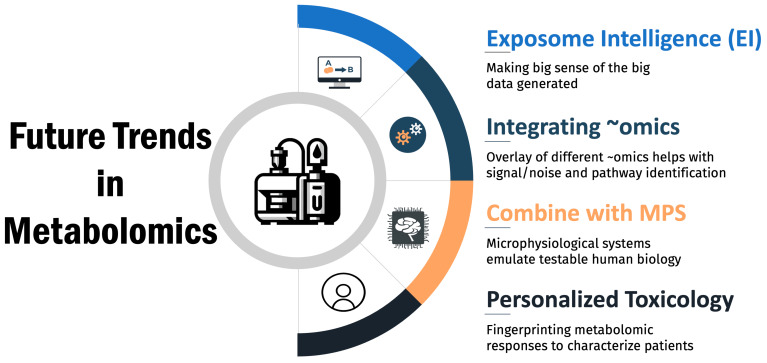
Future trends in metabolomics.

**Table 1 metabolites-14-00098-t001:** **Some examples of metabolomics used in toxicology.** This non-comprehensive list gives examples of studies successfully employing metabolomics. Δ = changed, ↓ = reduced, ↑ = increased.

Compound/Substance and Model	Mechanistic Insights	Biomarkers Identified	References
Acetaminophen (APAP) in rats	Mitochondrial oxidative and nitrosative stress; bile acid metabolism changes	cysteine-glutathione disulfide	[56,57,58,59]
Cigarette smoke in lung cells	Disruption of glycolysis, Krebs cycle, choline metabolism, and additive oxidative stress	glucose↓ lactate↑ Δ fumarate/malate	[60,61,62]
35 test substances in liver cells	Patterns of liver enzyme induction/inhibition, liver toxicity, and peroxisome proliferation	diverse	[63]
MPP^+^ in neurons	Dopaminergic neuron death pathways	diverse	[64]
Liver toxicants in rats	Early metabolomic changes	amino acids, bile acids↑;	[65]
Kidney toxicants in rats	Early metabolomic changes	Δ TCA cycle; urinary 2-oxoglutarate↑	[65,66]
Phenoxy herbicides in rats	Liver and kidney toxicity	Diverse pattern	[67]
2- and 3-aminopropanol in rats	Similarity of compounds allowing read-across	Diverse pattern	[68]
Spironolactone in fathead minnows	Changes in liver linked to declines in fecundity and other reproductive-related endpoints	Δ amino acid, tryptophan, and fatty acid metabolism	[3]
Dioxin-exposed humans vs. control	Distinct metabolite profiles	24 urinary steroid-related biomarkers	[69]
Tributyltin in zebrafish	Affected steroid biosynthesis metabolism	Diverse	[70,71]
6-propyl-2-thiouracil in zebrafish	(Neuro-) developmental toxicity	methionine↓, tyrosine↑, pipecolic acid↑ and lysophosphatidylcholine↑	[72]
Arecoline in rats	Δ lipid metabolism, amino acid metabolism, and vitamin metabolism	Δ D-Lysine, N4-Acetylaminobutanal, and L-Arginine	[73]

## Data Availability

No datasets were generated in this study.

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
