# Peer review of "Metabolomics in Preclinical Drug Safety Assessment: Current Status and Future Trends"

_metabolites, 2024, doi:10.3390/metabo14020098_

Round 1
Reviewer 1 Report
Comments and Suggestions for Authors
Hartung et al described current applications and future trends of metabolomics in toxicology and drug development. Metabolomics has emerged as a powerful technology for elucidating mechanisms of toxicity and improving preclinical drug safety assessment.
The review is very well written specially the introduction section. Although in section 4 they have discussed most critical points in a metabolomics study but they did this in a very short but not comprehensive way. It will be helpful if they can discuss the best possible ways to address those issues that they have mentioned in the section 4.
Additional point to consider. If possible it will be helpful if the author can add
1. A figure showing how exactly this metabolomics has emerged as a powerful technology for gaining insights into how human exposures impact metabolic pathways and influence disease outcomes as they have discussed in section 3. and
2. A table about the examples they discussed in section 5 "Role of Metabolomics in Toxicology" with some additional examples.
Reviewer 2 Report
Comments and Suggestions for Authors
Although the topic is interesting, this manuscript cannot be accepted in this form for publication.
Regarding the content:
The authors hardly address the place of metabolomics in preclinical drug safety assessment. However, they claim that this article aimed to provide an overview of the current state of metabolomics in toxicology and discusses its futures trends, focusing on its role in preclinical drug safety assessment.
Regarding the methodology:
The authors have to provide details on the literature search strategy and methodology (search equation, etc.).
Furthermore, the manuscript must be restructured in order to be clearer for the reader. Indeed, the sequence of the different sections does not seem to follow any logic.
Author Response
Reviewer 2
The authors hardly address the place of metabolomics in preclinical drug safety assessment. However, they claim that this article aimed to provide an overview of the current state of metabolomics in toxicology and discusses its futures trends, focusing on its role in preclinical drug safety assessment.
Indeed, we are not focusing too much on the current state of metabolomics in toxicology, which we have done in some of our earlier work cited:
- Bouhifd, M.; Hartung, T.; Hogberg, H.T.; Kleensang, A.; Zhao, L. Review: Toxicometabolomics. Journal of Applied Toxicology 2013, 33, 1365–1383, doi:https://doi.org/10.1002/jat.2874.
- Ramirez, T.; Daneshian, M.; Kamp, H.; Bois, F.Y.; Clench, M.R.; Coen, M.; Donley, B.; Fischer, S.M.; Ekman, D.R.; Fabian, E.; et al. Metabolomics in Toxicology and Preclinical Research. ALTEX 2013, 30, 209–225, doi:https://doi.org/10.14573/altex.2013.2.209.
- Bouhifd, M.; Beger, R.; Flynn, T.; Guo, L.; Harris, G.; Hogberg, H.T.; Kaddurah-Daouk, R.; Kamp, H.; Kleensang, A.; Maertens, A.; Odwin-DaCosta, S.; Pamies, D.; Robertson, D.; Smirnova, L.; Sun, J.; Zhao, L.; Hartung, T. Quality Assurance of Metabolomics. ALTEX 2015, 32, 319–326, doi:https://doi.org/10.14573/altex.1509161
- Maertens, A.; Bouhifd, M.; Zhao, L.; Odwin-DaCosta, S.; Kleensang, A.; Yager, J.D.; Hartung, T. Metabolomic Network Analysis of Estrogen-Stimulated MCF-7 Cells: A Comparison of Overrepresentation Analysis, Quantitative Enrichment Analysis and Pathway Analysis versus Metabolite Network Analysis. Archives of Toxicology 2016, 91, 217–230, doi:https://doi.org/10.1007/s00204-016-1695-x.
As stated in the abstract we discuss these and look out for future trends: “This review discusses current applications and future trends of metabolomics in toxicology and drug development.”
We agree that many examples are from areas of toxicology other than preclinical drug safety assessment; this was owed to the topic of the special issue but many pertinent examples in the literature come from chemical safety in general. If the editor feels more comfortable, we are happy to change the title to “Metabolomics in Preclinical Drug Safety Assessment: Current Status and Future Trends”.
Regarding the methodology:
The authors have to provide details on the literature search strategy and methodology (search equation, etc.).
This is not a systematic review. The authors are very familiar with this tool as Thomas Hartung holds a chair for evidence-based toxicology. A topic that broad is not suitable for this approach. The focus of future trends also does not lend itself to this methodology.
Furthermore, the manuscript must be restructured in order to be clearer for the reader. Indeed, the sequence of the different sections does not seem to follow any logic.
The structure of the article is:
Introduction
(goal: setting the scene)
Adverse Outcome Pathways vs. Pathways of Toxicity
(goal: pathway analysis as a primary goal of metabolomics)
We added: “A broader assessment of metabolites as enabled by metabolomics lends itself to the identification of affected pathways in toxicological research.”
Metabolomics to Understand Human Exposure Contributions to Disease and Treatment Efficacy
(goal: broader picture on metabolomics in biomedicine)
Metabolomics Data Quality Assurance in Preclinical Drug Safety Assessment
(goal: the central problem of metabolomics in general)
Role of Metabolomics in Toxicology
(goal: examples of use in toxicology)
State of the Art of Metabolomics in Toxicology
We removed Future Trends chapter title.
(goal: challenges of metabolomics in toxicology)
Future Trends for Metabolomics in Toxicology
(goal: current developments and expectations)
May we kindly ask the reviewer to advise on changes needed?
Reviewer 3 Report
Comments and Suggestions for Authors
The manuscript gives a comprehensive overview of the metabolomics and its importance in the preclinical drug safety assessments. The following are the concerns and suggestions to the authors.
Author should rephrase the sentences in the manuscript. For example, the last few lines of paragraph 1 in the introduction section.
Authors should add a pictorial description of adverse outcome pathway and pathways of toxicity.
Authors may add few more examples either in tabular format, of studies revealing how metaboloimics contributes to pre-clinical drug safety assessment.
Comments on the Quality of English LanguageLanguage need to be modified.
Author Response
Thanks for your review - see attached our replies
Thomas

Reviewer 4 Report
Comments and Suggestions for Authors
Interesting topic, the review fullfills its aim. The paper is not novel, but can be useful to busy readers to have information about this argument. The text is well written. However, I suggest minor corrections like no capital letter after colons, and reviewing all the acronyms.
Author Response
Reviewer 4
Interesting topic, the review fulfills its aim. The paper is not novel, but can be useful to busy readers to have information about this argument. The text is well written. However, I suggest minor corrections, like no capital letter after colons, and reviewing all the acronyms.
Thanks for the positive assessment. We screened the text for these errors.